# Analysis on outsourcing service behavior of rice pest and disease control based on Heckman selection model—A case study of ten counties in Fujian Province

**Liangmei Cai[1], Linping Wang[2]***

**1** College of Plant Protection, Fujian Agricultural and Forestry University, Fuzhou, China, **2** College of Economics and Management, Fujian Agricultural and Forestry University, Fuzhou, China

* linpingwang@fafu.edu.cn

**Data Availability Statement:** All relevant data are within the manuscript and its Supporting Information files.

## Abstract

Under the background of relatively slow agricultural labor transfer and land circulation, agricultural production outsourcing has become the main means of agricultural modernization. In order to provide a beneficial perspective for appropriately expanding the scale of rice control outsourcing services, we investigated the situation of rice control outsourcing in ten counties of Fujian Province, and analyzed the factors influencing rice farmers' decision-making and control degree by using Heckman model. First of all, the main factors affecting farmers' participation in outsourcing are agricultural labor force, whether family members are cooperative members, planting area, proportion of grain income, degree of organization of outsourcing team, region and so on. Secondly, agricultural labor force, cooperative members, planting area, part-time behavior, mechanical efficiency of prevention and control organization, and region are the main factors affecting the scale of control outsourcing. Thirdly, from a regional perspective, the rice farmers in northern and Western Fujian are more dependent on outsourcing services consumption compared with the rice farmers in Southern Fujian. These results have a clear impact on policymakers, indicating that policy and measures should encourage the prevention and control of the nature of cooperation, and improve the advanced nature of outsourcing facilities of plant protection equipment, thereby effectively improving the professional level of rice pest and disease control.

## 1. Introduction

Outsourcing is a manifestation of specialization. Its concept can be traced back to Adam Smith's understanding of specialization in the wealth of nations in the 17[th] century. At that time, due to the traditional manual labor as the main agricultural technology, and the scale is not enough, professional division of labor was mainly applied in industry. With the development of agricultural technology, some data showed that the outsourcing services of land preparation and harvest appeared in Africa and the United States from the 18[th] century to the 19[th] century [1,2]. However, until the beginning of the 20[th] century, some scholars still held a

**Funding:** Linping Wang is sponsored by the National Natural Science Foundation of China (project number: 71873035); http://www.nsfc.gov.cn/. The funders had no role in study design, data collection and analysis, decision to publish, or preparation of the manuscript.

**Competing interests:** The authors have declared that no competing interests exist.

cautious attitude towards agricultural production outsourcing services. According to Ruttan [3], small farmers only need small machinery to complete some tasks of agricultural production, and the labor market division is limited. On the contrary, other scholars believe that agricultural outsourcing is a more effective management strategy for small farms [4].

The research on agricultural outsourcing service at home and abroad mainly focuses on the theory and welfare of agricultural outsourcing service [5,6], the evaluation of outsourcing service participation in all aspects [7,8], and the estimation of outsourcing service production efficiency [4,9].

Agricultural production outsourcing services is developing rapidly in China, which has become one of the important ways to improve agricultural productivity, to promote the transformation of agricultural production mode and to ensure food safety. On the one hand, outsourcing services in production can replace household labor, make up for labor shortage caused by the transfer of non-agricultural sectors, and ensure the smooth progress of agricultural production [7]. On the other hand, outsourcing services can realize the scale operation of specific production links [8].

Some studies showed that the welfare effect of field management outsourcing is higher than that of other links [6]. However, compared with land preparation, seedling raising, transplanting, harvesting and other outsourcing links, the degree of farmers' participation in outsourcing service of rice pest control is the lowest [7]. Meanwhile, the response to the outsourcing service of rice pest and disease control is insufficient (Pests usually include all kinds of harmful organisms, including viruses, fungi, animals and so on. Diseases and insect pests in this paper mainly refer to insect pests and arthropod diseases.). Taking rice production in Fujian Province as an example, we studied the factors that affect farmers' response to rice control outsourcing services and the factors that affect the scale of rice comprehensive management.

The outsourcing service of pest control is the most "special" part of the whole agricultural outsourcing service. The control effect is directly related to the yield loss of farmers. Therefore, there is a high operational risk, which is the most "difficult" link compared with other production links. It puts forward higher requirements for the coordination of agricultural technology and agricultural machinery.

Through the answers to the above questions, the contribution of this paper is to clarify the crux of the impact on the response and scale of agricultural outsourcing services, improve farmers' participation in the outsourcing service of rice pest control, and then integrate a series of outsourcing services such as seedling raising, cultivated land, prevention and harvest, so as to comprehensively improve the overall efficiency of agricultural outsourcing services. Expanding outsourcing means that agricultural development will break through the rigid thinking field of realizing scale operation only relying on the aggregation of agricultural land property rights, and realize the scale operation of prevention and control services through the aggregation of field management, so that farmers can participate in agricultural transactions and share the economy of division of labor. This is an important exploration to change the mode of agricultural production, to meet the demand of full flow of agricultural production factors and to improve agricultural labor efficiency.

The rest of the study will be arranged as follows. The second part is literature review. The third section introduces theoretical analysis, summarizes the data set, and demonstrates the regression analysis. The fourth part explains the empirical results. Finally, the fifth section will give the conclusion and discussion.

## 2. Literature review

Agricultural production outsourcing service is the behavior of outsourcing some or all links in the process of agricultural production to large producers, professional service teams or agricultural cooperatives [10]. As of January 2021, there are 93,000 prevention and control institutions in China, and the outsourcing service coverage rate of rice pest control of the three major crops has reached 41.9% (Data from the Ministry of Agriculture and Rural Affairs of The People's Republic of China http://www.moa.gov.cn/xw/zwdt/202101/t20210117_6360031.htm). The objects of prevention and control outsourcing services extend from the three major crops to other crops. The existing research mainly focuses on the theoretical basis of prevention and control outsourcing, the factors affecting the participation of agricultural outsourcing, and the impact of agricultural outsourcing services on other production factors.

### 2.1 Theoretical basis of agricultural outsourcing service

Agricultural outsourcing service is regarded as a form of agricultural division of labor, which comes from the progress of technology. Through the third-party outsourcing service organization, we can aggregate the field labor demand of a single agricultural production link, and use more advanced agricultural equipment instead of manual operations to realize large-scale production. Many scholars usually use Smith's division of labor theory to explain agricultural outsourcing services [11–13]. They believe that market capacity is the key to promoting the development of agricultural outsourcing service industry.

The market capacity includes two levels. One is the density of individual participation of farmers. Since farmers are the main decision-makers in choosing agricultural outsourcing services, the degree of outsourcing adoption is directly related to the aggregation degree of decision-makers. The higher the degree of participation in outsourcing, the larger the scale of the outsourcing market [14]. The second is the number of outsourcing service transactions after the summary of outsourcing decisions. The assessment of the tradability and quality supervision of agricultural activities increases the frequency of agricultural operations in the vertical division of labor [11]. The expanding and deepening of agricultural division of labor creates conditions for the sustainable development of agricultural outsourcing services.

### 2.2 Study on the influencing factors of agricultural outsourcing services participation

Many scholars paid attention to the influence of external environment on the choice of outsourcing services, such as the influence of farmland ownership on agricultural machinery outsourcing [15], and studied the relationship between agricultural outsourcing services and land transfer policies [16], as well as the influence of prevention and control machinery subsidy policies on the adoption of prevention and control outsourcing [17]. Wang Zhi-gang [18] analyzed the influencing factors of rice farmers using agricultural producer services through logit model. He believes that, compared with the realization of agricultural scale operation through land circulation, it is easier to realize social services through production links. Li Qiao and Zhang Bo [19] believe that the age of farmers, business scale, net income per capita, agricultural production and regional location have a significant impact on the demand for agricultural outsourcing services. The research of Gillespie *et al.* [20] and Cai Jian *et al.* [21] also confirmed the similar conclusion, indicating that family labor characteristics, planting scale, business characteristics and other factors can affect agricultural outsourcing behavior.

## 2.3 The influence of agricultural outsourcing services on other production factors

First, the impact of agricultural outsourcing services on land elements. Some scholars believe that agricultural outsourcing services can help farmers expand the planting area and alleviate the negative impact of labor loss on the planting area [22]. However, Chen Chao and Tang Ruo-di [23] analyzed the impact of agricultural production outsourcing on the scale of grain farmers' renting farmland from the perspectives of the heterogeneity of farming scale and production capacity, and found that agricultural production outsourcing services did promote the large grain farmers' renting farmland, while inhibiting the expansion of small farms. Jiang Song *et al.* [24] empirically tested the impact degree and direction of agricultural outsourcing services on land moderate scale operation based on chip data, and found that different types of outsourcing services have different impacts on land moderate scale operation, such as irrigation services, machine farming services, pest control services and planting planning services have significantly positive impacts on land moderate scale operation. The marginal influence coefficient decreases in turn. However, the impact of agricultural outsourcing on productive purchasing services and labor organization is not significant. Luo Bi-liang [25] conducted an empirical analysis on the sample survey data of 2704 farmers in nine provinces and regions of China, and concluded that agricultural outsourcing can significantly reduce farmers' abandonment of agricultural land, while the development of agricultural outsourcing service market can significantly reduce the proportion of agricultural land abandonment caused by land fragmentation.

Secondly, the impact of agricultural outsourcing services on Farmers' income. Chen Hong-wei, Mu Yue-ying [26] and Yang Zhi-hai [27] used the endogenous transformation model to draw the conclusion that agricultural productive services have a significant impact on the growth of farmers' income. Among them, the welfare effect of Yang Zhi-hai's [27] field control outsourcing is higher than that of other production links, while the welfare effect of sowing outsourcing is not significant.

## 2.4 Research gap

The above scholars have done fruitful research on agricultural outsourcing service production, and these results will provide a useful reference for the further research of this paper. Some scholars began to pay attention to the impact of outsourcing services on crop production in the process of prevention and control. Ying Rui-yao and Xu Bin [28] used PSM model to study the impact of rice outsourcing services on pesticide application intensity, and concluded that outsourcing services reduced the application frequency of pest control under the premise of ensuring yield; Chen Pin *et al.* [29] verified that the use of prevention and control outsourcing services by farmers can make up for production losses in a timely manner.

However, the existing research results may have the following limitations: first, the empirical analysis of prevention and control outsourcing services based on Farmers' micro survey is relatively lack. Some scholars found that there were significant differences in the participation of agricultural outsourcing services [7], but there was no in-depth explanation for the low participation of prevention and control outsourcing services; Secondly, all prevention and control outsourcing services are regarded as homogeneous service products, and the causal effects of the heterogeneity of geographical conditions and production Endowments on the adoption of prevention and control outsourcing are not fully investigated. Only by gathering the needs of multiple farmers and forming the service market capacity, can the important role of agricultural outsourcing service be played [11]. Therefore, it is necessary to further analyze on the behavior of farmers using prevention and control outsourcing in the next chapter.

## 3. Materials and methods

### 3.1 Theoretical analysis

The outsourcing service of rice pest control can be used as a choice for growers. This paper will focus on an agricultural technology adoption decision model. Under the framework of economic theory, the basis of agricultural technology adoption is expected utility maximization or expected profit maximization. Early studies focused on comparing the marginal benefits and costs of new technologies. Kong Xiang-zhi [30] believes that even if the expected marginal income is greater than zero, it is not necessarily the best choice for producers. He stressed that the key to technology adoption lies in the comparison between the expected net income of technology and the net income of original technology. Only when the former is greater than the latter, will producers choose new technology. The different risk premium of producers will affect the allocation of production resources and the decision of production. Koundouri, Nauges *et al.* [31] believe that people's attention to the adoption of any agricultural technology mainly comes from two aspects: (1) the perceived risk of future agricultural output after the adoption; (2) the product or cost risk related to agriculture itself [31]. Based on the theoretical model of Koundouri, Nauges and others [31], this paper deduces the conditions for farmers to take preventive and control outsourcing services:

Firstly, the expression of the expected utility maximization of rice growers' income is established.

$$\max E[U(\Pi)] = \max \int [U(p \cdot f(h(\partial) \cdot T \cdot X_C, X_{-C}) - rm]$$

In this expression, p represents the output price, and f(.) is the production function of unit output. In f(.), $X_C$ is the prevention and control method of input. Considering the different selection methods of different farmers, the coefficient h($\partial$) represents the endowment characteristics of farmers. At the same time, the author believes that the control of diseases and insect pests is directly related to the yield. The level of different control technologies will affect production. T is the technical level, r is the unit production cost, m is the planting scale, and $X_{-C}$ is the input of production factors other than control technology. Thus, the expected utility of traditional control is expressed as follows:

$$\max E[U(\Pi^0)] = \int [U(p \cdot f(h(\partial) \cdot T \cdot X_c^0, X_{-c}^0) - r_0 m]$$

Finally, the expected utility of outsourcing is expressed as follows:

$$\max E[U(\Pi^1)] = \int [U(p \cdot f(h(\partial) \cdot T \cdot X_c^1, X_{-c}^1) - r_1 m]$$

Only when $\max E[U(\Pi^1)] > \max E[U(\Pi^0)]$, will the growers choose prevention and control outsourcing services. In this formula, it can be seen that whether farmers choose prevention and control outsourcing service depends on endowment characteristics, production characteristics, planting scale and outsourcing organizational factors of farmers.

### 3.2 Data source and sampling characteristic

**3.2.1 Data source.**   Stratified sampling was used in current study. Two factors were considered in the selection of survey area: One is to select the high-yield and planting areas of rice in Fujian Province according to the statistical yearbook; Second, according to the "Opinions on the Implementation of Rice Production Functional Areas", it is stipulated that the permanent basic farmland management shall be implemented within the scope of rice production functional areas, and no unit or individual shall occupy or change the purpose without authorization. The delineation of rice functional area is related to the national "grain bag". Therefore,

**Table 1. Survey area of rice pest control in Fujian Province.**

| City | County | Town | |
|------|--------|------|------|
| Nanping | Shaowu | Hongdun | Nakou |
| | Jianyang | Jiangkou | Chongluo |
| Sanming | Jianning | Lixin | Xikou |
| | Youxi | Yangzhong | Xiwei |
| Longyan | Shanghan | Zhongdu | Lufeng |
| | Zhangting | Guanqian | Tongfang |
| Zhangzhou | Zhangpu | Chihu | Shiliu |
| | Longhai | Dongsi | Jiuhu |
| Quanzhou | Yongchun | Dapu | Penghu |
| | Nanan | Matou | Yingdu |

it is necessary to select the area to be listed in the rice production functional area catalogue. In this paper, 10 counties were selected as the survey area, with 2 towns in each county and 2 villages in each town. A total of 40 villages (**Table 1**) were selected to ensure the representativeness of rice planting in the survey area.

**3.2.2 Population sample characteristics.** From 2018 to 2019, the surveys on rice control in the above 40 villages were completed. The questionnaire was distributed to 15 households in each village (12, 13 or 14 households in some villages). A total of 600 questionnaires were distributed, 537 of which were valid, with an effective rate of 89.5%. The overall sample is as follows: (1) From the perspective of farmers' characteristics, most of the respondents are middle-aged people, with an average age of 52 years, an average education of 7 years and an average farming life of 30 years. 43% of the farmers are engaged in non-agricultural work. (2) From the perspective of family characteristics, the average agricultural labor force accounts for 56% of the family population, 38% of the family members participate in the agricultural cooperative, and 8% of the families are village cadres. (3) The production and management characteristics of farmers are as follows: the average planting area is 31.06 mu (One mu equals 1/15 ha.), 60% of the families grow rice for the purpose of sales, and the average income from rice planting accounts for 55% of the family income. (4) In 537 survey samples, 203 households participated in the outsourcing service of rice pest control, accounting for 37.8% of the total.

## 3.3 Model setting

The Heckman selection model was proposed by James J. Heckman who won the Nobel Prize in economics in 2000. In order to solve the interference of self-selection in the process of decision-making, it is used by many scholars as an effective tool for many scholars to analyze the behavior and decision-making of subjects. The decision-making of subject behavior includes two parts. The first part is the subject's will, attitude and response to a certain behavior, and the second part usually reflects the subject's degree of implementation. At present, many scholars are using Heckman selection model to study the adoption of agricultural technology mode and people's livelihood consumption decision [32,33]. Farmers' choice of outsourcing control can also be seen as the combination of two-stage decision-making process. In the first stage, rice growers respond to whether they participate in outsourcing service of rice pest control; The second stage is to determine the pest control area. Moreover, it is not a random decision of the growers. There are some unobservable factors, such as labor ability, which will also affect the decision-making of the outsourcing. In other words, as an independent variable, decision-making has endogenous problems caused by self-selection. Therefore, this paper uses Heckman two-stage model to analyze the influencing factors of outsourcing service response and

control scale in rice pest control decision-making, in order to reduce selective bias and improve the credibility of the conclusion.

**3.3.1 Heckman selection model setting.** There are two stages in Heckman's selection formula. The first stage is whether the rice planters will participate in outsourcing service of rice pest control. The second stage is to decide the outsourcing service area of rice pest control on the basis of the first stage. It includes a selection equation of $y_1$ and a related result equation of $y_2$. Only when $y_1^* > 0$, can $y_2$ be observed. If $y_1^* \leq 0$, $y_2$ does not take a value. The logic is as follows:

Firstly, the response model (1) was established, and the response probability equation was constructed with whether outsourcing control is selected as the dependent variable. In order to determine the influencing factors of rice farmers' participation in outsourcing control behavior, Probit estimation was conducted for all 537 samples using the following influencing factors:

$$y_i = \beta x_i + \varepsilon_i \ (i = 1, 2, 3 \ldots \text{n}) \tag{1}$$

$y_i$ is a binary discrete variable. $y_i = 0$ means that the growers choose to control the pests independently, and $y_i = 1$ means that the growers choose to outsource the pest control. $x_i$ is a vector composed of exogenous variables that determine farmers' choice of prevention and control behavior, including the farmers' characteristics, family characteristics, outsourcing prevention and control characteristics, external environment characteristics and other variables. β is the correlation coefficient, and $\varepsilon_i$ is the error term, which obeys normal distribution $\varepsilon_i \sim N (0, \sigma^2)$.

Then, the decision-making model of outsourcing service scale of rice pest control was established, and the response intensity of growers was estimated by using Heckman model. By using the relevant independent variables, this paper made regression analysis on 537 samples of rural household outsourcing prevention and control, and finds out the factors influencing the scale of rural household outsourcing prevention and control. Considering that the second stage OLS regression may have selective bias, the nonlinear term can be ignored, and the correlation error is assumed to be joint normal distribution and homogeneous variance, the maximum likelihood estimation is a more effective method.

The decision-making model of outsourcing scale and degree of rice farmers' pest control was established.

$$Y_m = \alpha Z_i + \mu_i \tag{2}$$

In formula (2), $Y_m$ represents the scale of farmers' outsourcing. $Z_i$ is the explanatory variable that affect farmers' decision on outsourcing services, including the endowment of growers, family characteristics, production characteristics, etc. α is the coefficient to be estimated, $\mu_i$ is random error term. The list of variables needs to be specified in the selection equation and result equation of Heckman model. In addition, it should be noted that if the model identification is based on the nonlinear functional form, the same set of explanatory variables can be used in the two equations [34]. In order to more robust identification, we can also add the exclusion constraint variable, which is the exogenous variable that appears in the selection equation and excluded from the result equation.

Finally, use the likelihood ratio test (LR test) verify whether the sample has selectivity bias. If the correlation between the error items is significant, then the existence of selective bias is proved. The Heckman two-stage model is effective.

**3.3.2 Selection and description of model variables.** Previous studies have shown that agricultural technology decision-making adoption may be affected by factors such as farmers' endowment, labor force characteristics and external environment [35,36], In particular, the

**Table 2. The definition and descriptive statistics of farmers' response variables to control outsourcing.**

| Variable name | Variable definition | Mean | Standard deviation | Expected impact |
|---|---|---|---|---|
| **Independent variables** | | | | |
| **Characteristic variables of farmers** | | | | |
| Age | Actual age of decision maker (unit: year) | 52.12 | 9.82 | uncertain |
| Farming experience | Decision maker's years of farming (unit: year) | 29.93 | 12.89 | uncertain |
| Education level | Years of Education (unit: year) | 6.99 | 3.23 | uncertain |
| Part-time Farming | Whether there is Part-time Farming | 0.43 | 0.50 | uncertain |
| **Family characteristic variables** | | | | |
| Agricultural labor force | Number of labor input per unit area | 0.87 | 1.58 | negative |
| Member of a cooperative | Not cooperative member = 0, cooperative member = 1 | 0.38 | 0.49 | positive |
| **Production and operation characteristic variables** | | | | |
| Planting scale | Rice planting area (unit: mu) | 31.06 | 66.91 | positive |
| Proportion of planting income | Proportion of rice planting income in household income | 0.55 | 0.36 | positive |
| planting distribution | Different from the crops in the adjacent plots = 0, Same as crops in adjacent plots = 1 | 0.79 | 0.41 | positive |
| Land conditions | Difficulty degree of working land conditions for plant protection machinery Easy = 1, General = 2, Difficult = 3 | 1.51 | 0.64 | uncertain |
| **Outsourcing organizational variables** | | | | |
| Efficiency substitution of outsourcing machinery prevention | Comparison of efficiency between outsourcing organization and self-owned plant protection machinery: Outsourcing control efficiency is higher than self-owned control efficiency = 1, otherwise = 0 | 0.37 | 0.48 | positive |
| Outsourcing prevention price | Unit area outsourcing control charge (unit: yuan per mu) | 11.68 | 19.07 | negative |
| Organization degree of rice regional production | Whether the scope of outsourcing service covers the village: No = 0, Yes = 1 | 0.76 | 0.43 | positive |
| **External environment variables** | | | | |
| Regional variables | South Fujian = 1, North Fujian = 2, West Fujian = 3 | 1.86 | 0.74 | uncertain |

labor productivity gap of specialized suppliers will affect the probability of farmers participating in agricultural production outsourcing services [37]. At the same time, the characteristics of labor force and the degree of land fragmentation are also the key factors affecting the socialized service of agricultural machinery [38]. In this paper, the response of outsourcing service of rice pest and disease control and the scale of outsourcing control were taken as the explanatory variables of selection equation and result equation respectively. The participation of outsourcing service of rice pest and disease control comes from the questionnaire "whether rice growers will use outsourcing service of rice pest and disease control in 2018" in the questionnaire of Fujian Province. The scale of outsourcing service of rice pest and disease control is "the area of rice growers' outsourcing service of rice pest and disease control in 2018" × "the number of controls", and the average outsourcing scale is 185.89 mu.

On the basis of the existing research results and field research, this paper considers that the selection and scale of outsourcing service of rice pest and disease control mainly depend on the characteristics of householder, family characteristics, production and management characteristics, outsourcing services and external environment variables, which are further subdivided into 13 indicators (Table 2).

*(1) Characteristic variables of farmers.* The characteristic variables of farmers include age, farming experience, education level and part-time employment behavior. Generally speaking, age will affect the ability to accept new things, and at the same time, the awareness of risk aversion is stronger. The older the expected growers are, the negative impact on the participation of prevention and control outsourcing. However, some scholars believe that the older the person is, the worse the physical strength is, and may not be competent for agricultural work [27].

The elderly has an internal demand for outsourcing prevention and control services, and "outsourcing" is used to reduce the field management time. Therefore, the impact of age on the outsourcing of prevention and control services is uncertain. Farmers with more planting experience have higher ability to learn agricultural knowledge and skills, and are more likely to accept new pest and disease control modes. However, some scholars [39] believe that path dependence will occur after planting experience accumulates to a certain extent. Farmers will strengthen their own planting mode in later practice, and their willingness to accept new control services will be weakened. Therefore, the impact of planting experience on outsourcing prevention and control services remains to be determined.

In addition, the higher the education years of the growers, the more likely they are to go out to work. On the one hand, if the rice management decision-makers get more non-agricultural employment opportunities, then the investment in field management time is relatively reduced, and they may choose the method of prevention and control outsourcing to save the farmland investment time; on the other hand, it may be weakened by the increase of non-agricultural employment income so as to reduce the investment in the prevention and control of rice diseases and insect pests. Therefore, the influence of education level and part-time business behavior on outsourcing service is uncertain.

*(2) Family characteristic variables*. Family characteristic variables include agricultural labor force and whether family members are cooperative members. Generally speaking, the less labor input per unit area, the heavier the farming tasks that the family members with labor capacity have to undertake, and the more likely they are to use prevention and control outsourcing to save manpower, so farmers are more willing to participate. If the family members are cooperative members, it means that there are more channels to obtain agricultural production technology information, and they are more willing to accept the new prevention and control service mode.

*(3) Production and operation characteristic variables*. The characteristics of production and management include planting area, proportion of rice planting income, planting layout, and land conditions of plant protection machinery operation. The proportion of rice planting area and planting income shows the degree of family dependence on rice production. The higher the dependence on rice production, the stronger the willingness to adopt the new outsourcing control mode to improve the efficiency of pest and disease control In the planting layout, the crops in adjacent plots are consistent to form a production scale, which has a positive impact on rice control outsourcing.

If it is more difficult for the plant protection machinery to enter the land during the control operation, farmers may be willing to outsource the field with complex terrain to reduce their labor intensity. If the land condition has exceeded the tolerance of the machinery held by the outsourcing organization to the land requirements, then the response demand can not be realized. On the other hand, some prevention and control teams use plant protection Unmanned Aerial Vehicle to overcome terrain operation obstacles. Therefore, the impact of land condition difficulty on the participation and scale of outsourcing prevention and control services remains to be determined.

*(4) Outsourcing prevention service variables*. The outsourcing service variables include the efficiency substitution of prevention and control machinery, the price of prevention and control, and the organization degree of prevention and control team. Compared with the farmer owned plant protection machinery, the higher the efficiency of the control machinery equipped by the outsourcing organization, the more obvious the comparative advantage of the latter, and the stronger enthusiasm for purchasing outsourcing prevention and control services. The lower the price of outsourcing prevention and control services, the higher the enthusiasm of participation. The organization degree of rice regional production is expressed by

"whether the service scope of prevention and control organization covers the village". It is assumed that the higher the degree of organization of rice regional production, the higher the willingness of farmers to choose outsourcing.

*(5) External environment variables*. The external environment variables are regional variables. Due to the differences in the publicity and administrative measures of promoting social prevention and control services in different regions, regional dummy variables are set to compare the participation of different regions in outsourcing service of rice pest and disease control According to the setting of explanatory variables, the definition and descriptive statistics of explanatory variables are given (Table 2).

## 4. Model estimation results

Stata was used to calculate the regression results of outsourcing control decision based on Heckman selection model. Wald chi2 (12) = 1316.68, prob > chi2 = 0.0000 in the model, which was significant at 1% level, indicating that the overall fitting effect of the model was good. The p value of athrho is 0.000, which indicates that the estimated value of athrho is significantly different from 0 at 1% confidence interval, so there is sample selection bias. At the same time, the LR test results of the two-stage error correlation test are also significant at the 1% level, which shows that the Heckman selection model is effective and reasonable (Table 3).

### (1) Effects of farmers' characteristics on response of outsourcing services and scale of outsourcing service of rice pest and disease control

In terms of the characteristics of farmers, the response of part-time business behavior to outsourcing services did not pass the significance test, and the coefficient was positive, indicating that part-time business behavior can promote the enthusiasm of rice farmers to purchase outsourcing prevention and outsourcing service of rice pest and disease control, but it is not the main factor affecting the response of outsourcing services.

In addition, part-time business behavior has a significant impact on the scale of outsourcing services, which has passed the 5% level significance test and the coefficient is negative. This means that compared with farmers without part-time employment, the scale of outsourcing service of rice pest and disease control of farmers with part-time business is smaller. The possible explanation is that in the rice production season, it is generally necessary to apply the pesticide 3–4 times, and the outsourcing team can obtain the control scale through this circuitous service. However, in the actual process of outsourcing pesticide application, there is a risk of supervision on the control effect. The part of farmers who have part-time business behavior need to travel between the part-time business location and the field site, resulting in the relatively high cost of on-site supervision and the relatively small flexibility of supervision time, forcing such farmers to reduce the number of control outsourcing. Therefore, part-time farmers have a significant negative impact on the scale of outsourcing services.

### (2) Influence of family characteristics on response and the scale of outsourcing service of rice pest and disease control

In terms of family characteristics, family members who are cooperative members have a significant impact on the participation of outsourcing service of rice pest and disease control, which has passed the 1% level significance test and the coefficient is positive, which is the same as expected. In the survey of Fujian Province, more than 80% of the existing outsourcing service of rice pest and disease control organizations come from the agricultural and Agricultural Machinery Cooperatives in this town. Among the family members, there are cooperative

**Table 3. The regression results of outsourcing prevention decision based on Heckman model.**

| Heckman selection model | | | | | | |
|---|---|---|---|---|---|---|
| | | | | | | Number of obs = 537 |
| | | | | | | Censored obs = 334 |
| | | | | | | Uncensored obs = 203 |
| Log likelihood = -1425.407 | | | | | | |
| | | | | | | Wald chi2(12) = 1316.68 |
| | | | | | | Prob > chi2 = 0.0000 |

| The scale of outsourcing service of rice pest and disease control | | | | | | |
|---|---|---|---|---|---|---|
| | Coef. | Std. Err. | z | P>\|z\| | [95% Conf. Interval] | |
| age | -.2658 | 1.3783 | -0.19 | 0.847 | -2.9673 | 2.4357 |
| exp | .3895 | 1.0339 | 0.38 | 0.706 | -1.6369 | 2.4158 |
| Part-time | -36.6673** | 17.2345 | -2.13 | 0.033 | -70.4463 | -2.8883 |
| labor | 23.8880** | 11.0097 | 2.17 | 0.030 | 2.3094 | 45.4665 |
| size | 2.9222*** | .0892 | 32.78 | 0.000 | 2.7475 | 3.0969 |
| member | -42.4429** | 18.2357 | -2.33 | 0.020 | -78.1844 | -6.7015 |
| **Land conditions** | | | | | | |
| general | 10.9097 | 16.2555 | 0.67 | 0.502 | -20.9505 | 42.7698 |
| difficult | 24.8768 | 29.5609 | 0.84 | 0.400 | -33.0616 | 82.8152 |
| Proportion of planting income | -.1613 | .2627 | -0.61 | 0.539 | -.6761 | .3536 |
| **Regional variables** | | | | | | |
| North Fujian | 119.1083*** | 18.6751 | 6.38 | 0.000 | 82.5059 | 155.7108 |
| West Fujian | 52.6125 ** | 23.2765 | 2.26 | 0.024 | 6.9915 | 98.2336 |
| Efficiency substitution | 67.0120* | 37.1099 | 1.81 | 0.071 | -5.7222 | 139.7461 |
| _cons | -14.0824 | 72.2447 | -0.19 | 0.845 | -155.6793 | 127.5145 |

| Whether to participate in outsourcing service of rice pest and disease control | | | | | | |
|---|---|---|---|---|---|---|
| | Coef. | Std. Err. | z | P>\|z\| | [95% Conf. Interval] | |
| age | -.0086 | .0133 | -0.64 | 0.520 | -.0347 | .0175 |
| Education level | -.0212 | .0216 | -0.98 | 0.326 | -.0635 | .0211 |
| exp | .0003 | .0101 | 0.02 | 0.980 | -.0196 | .0201 |
| Part-time | .0376 | .1584 | 0.24 | 0.812 | -.2729 | .3482 |
| labor | -.2897*** | .0988 | -2.93 | 0.003 | -.4833 | -.0960 |
| size | .0048*** | .0014 | 3.36 | 0.001 | .0020 | .0077 |
| Member | .8892*** | .1492 | 5.96 | 0.000 | .5968 | 1.1816 |
| planting distribution | .1743 | .1622 | 1.07 | 0.282 | -.1435 | .4922 |
| price | .0511 | .0428 | 1.20 | 0.232 | -.0327 | .1349 |
| **Land conditions** | | | | | | |
| general | -.2260 | .1515786 | -1.49 | 0.136 | -.5231 | .0711 |
| difficult | -.1418 | .2745 | -0.52 | 0.605 | -.6799 | .3963 |
| Proportion of planting income | .0039* | .0030 | 1.68 | 0.093 | -.0006 | .0084 |
| **Regional variables** | | | | | | |
| North Fujian | -.5061** | .2006 | -2.52 | 0.012 | -.8992 | -.1130 |
| West Fujian | -.0636 | .2240 | -0.28 | 0.777 | -.5026 | .3754 |
| Organization degree | 1.2499*** | .2921 | 4.28 | 0.000 | .6774 | 1.8225 |
| _cons | -1.3678 | .6660 | -2.05 | 0.040 | -2.6732 | -.0624 |
| /athrho | -1.0988 | .2087 | -5.26 | 0.000 | -1.5079 | -.6897 |
| /lnsigma | 4.6510 | .0792 | 58.71 | 0.000 | 4.4957 | 4.8063 |
| rho | -.8001 | .0751 | | | -.9066 | -.5978 |
| sigma | 104.6891 | 8.2940 | | | 89.6324 | 122.275 |

(*Continued*)

**Table 3.** (Continued)

| lambda | -83.7586 | 13.6739 | | | -110.5589 | -56.9583 |
|---|---|---|---|---|---|---|
| LR test of indep. Eqns. (rho = 0): chi2(1) = 9.90 Prob > chi2 = 0.0016 | | | | | | |

Note

***, **, *significant at the 1%, 5%, 10% level.

members, which can simplify the transmission channel of agricultural control information, and at the same time, it can virtually strengthen the collective decision-making behavior of prevention and control, save the transaction costs of outsourcing control, and make these farmers more willing to participate in the outsourcing control. However, the effect of family members as cooperative members on the scale of prevention and control passed the 5% level significance test and the coefficient was negative. This means that if family members come from cooperatives, it will have a significant negative impact on the scale of outsourcing service of rice pest and disease control. The possible reason is that the farmers with cooperative background can obtain the pest forecast and control information more quickly, grasp the best control period, and reduce the purchase times of rice outsourcing services, so it has obvious negative impact on the scale of outsourcing service of rice pest and disease control.

The agricultural labor force has a significant impact on the participation in outsourcing service of rice pest and disease control. It has passed the 1% level significance test and the coefficient is negative, which is the same as expected. Outsourcing control services solve the field labor consumption, especially in the case of insufficient manpower, the smaller the ratio of household labor per unit area of farmland means that the field management time is insufficient, and more active participation in outsourcing service of rice pest and disease control. Moreover, this factor has a significant impact on the scale of outsourcing service of rice pest and disease control. Through the 5% level significance test and the coefficient is positive, it shows that agricultural labor input can promote the formation of outsourcing service of rice pest and disease control scale. The possible reason is that for the farmers who have chosen to outsource the prevention and control behavior, the greater the labor input, the stronger the dependence on agriculture, and they may be more concerned about the yield loss caused by diseases and insect pests, and are willing to increase the number of outsourcing services in order to avoid the yield risk caused by diseases and insect pests, so the scale of outsourcing control has a positive impact.

### (3) Effects of production and management characteristics on response and the scale of outsourcing service of rice pest and disease control

In terms of production and management characteristics, the planting area had a significant positive impact on the response and scale of outsourcing service of rice pest and disease control, which passed the 1% level significance test and the coefficient was positive, which was the same as expected. It shows that the planting area has a certain scale, and it is easier to accept new agricultural technology and new mode to improve the efficiency of farmland production.

In addition, the significance test of the proportion of rice planting income on the response of outsourcing service of rice pest and disease control passed the 10% level significance test, and the coefficient was positive, indicating that the proportion of rice planting income had a positive impact on the response of outsourcing service of rice pest and disease control, which was consistent with the expectation in the paper. However, the results of the model show that the proportion of rice planting income has a significant negative impact on the scale of

outsourcing service of rice pest and disease control. The possible reason is that farmers with larger proportion of rice planting income tend to increase agricultural input, and their families have advanced nature of plant protection machinery. Only when pests and diseases are in serious danger, can they buy outsourcing services sporadically. In the process of prevention and control, because of the existence of sunk cost, it can't completely rely on outsourcing prevention and control, so it has a negative impact on the formation of outsourcing prevention and control scale.

### (4) Effects of outsourcing service characteristics on response and control scale of outsourcing service of rice pest and disease control

In terms of the characteristics of the outsourcing services, the degree of organization of the control team had a significant positive impact on the response of outsourcing service of rice pest and disease control, which passed the 1% level significance test and the coefficient was positive, which was the same as expected. It shows that the larger the scope of organizational services, the more likely to increase the probability of participating in outsourcing. In fact, the service scope of outsourcing organization represents the availability of outsourcing service information for rice farmers. The wide range of services means that the lower the transmission cost of outsourcing prevention and control information, the lower the transaction cost of farmers and outsourcing organizations to discuss prevention and control matters. Meanwhile, rice farmers with more outsourcing technology information channels are more vulnerable to the new outsourcing prevention and control service mode.

It should be noted that the price of outsourcing prevention and control in the model is not significant, and theoretically the price of prevention and control is closely related to the participation of outsourcing. However, in the survey of prevention and control outsourcing in Fujian Province, the price of prevention and control in different regions of Fujian Province is relatively fixed. The reason is that the provincial administrative department issued the guidance price for outsourcing service of rice pest and disease control, and then the Municipal Agricultural Bureau proposed the prevention and control guidance price suitable for the local planting and production situation according to the actual situation, and carried out strict management on the outsourcing organization. Secondly, in recent three years, Fujian Agricultural Department has allocated special outsourcing subsidy fund every year, which is 20% - 30% of the outsourcing price. According to the author's research, more than 70% of the outsourcing farmers enjoy the government's outsourcing prevention and control subsidy. In general, the price difference of outsourcing service of rice pest and disease control is not significant, so the price of outsourcing service of rice pest and disease control is not significant. In addition, the efficiency of outsourcing prevention and control machinery passed the significance test of 10% level and the coefficient was positive, indicating that the efficiency advantage of outsourcing prevention and control machinery was conducive to the formation of outsourcing service prevention scale, which was the same as expected.

### (5) Effects of external environment characteristics on response and scale of outsourcing service of rice pest and disease control

In terms of external environmental characteristics, the impact of Northern Fujian on the response of outsourcing control services passed 5% significance test and the coefficient was negative, indicating that the response degree of rice farmers participating in outsourcing in Northern Fujian was not as good as that in Southern Fujian. Moreover, the influence of northern and Western Fujian on the scale of prevention and control has passed the 1% significance test and the coefficient is positive, which indicates that compared with southern Fujian, the

farmers in northern and Western Fujian have purchased more prevention and control services, and their outsourcing prevention and control behavior has changed from sporadic purchase to "prevention and control trusteeship" habit. In other words, the rice farmers in northern and Western Fujian are more dependent on the outsourcing service mode, which is conducive to the formation of division of labor and specialization of outsourcing service of rice pest and disease control.

## 5. Conclusion and suggestion

Based on the micro survey data of 537 rice growers in Fujian Province, this study uses the Heckman two-stage selection model to empirically investigate the influencing factors of outsourcing control participation behavior and outsourcing control scale. The main conclusions are as follows:

1. Generally speaking, the proportion of farmers participating in outsourcing service of rice pest and disease control is not high, accounting for 37.8% of the sample size; in the outsourcing control group, about 56.16% of the farmers join the cooperatives; in terms of the mechanical selection of the outsourcing group, about 63.55% of the farmers in the outsourcing group mainly focus on the UAV control of plant protection.

2. Agricultural labor force, whether family members are cooperative members, planting area, proportion of grain income, service scope of prevention and control organization and region are the main factors influencing rice farmers' participation in outsourcing service of rice pest and disease control. Especially, joining cooperatives and expanding service scope of prevention and control organizations can significantly improve the enthusiasm of farmers to participate in outsourcing service of rice pest and disease control. In addition, agricultural labor force, cooperative members, planting area, part-time farming behavior, mechanical efficiency of control organization and region are the main factors affecting the scale of rice farmers' outsourcing service of rice pest and disease control. Among them, the number of purchasing outsourcing services in northern and Western Fujian is relatively more, and the awareness of "prevention and control trusteeship" is stronger.

Agricultural outsourcing control, with more advanced control machinery and specialized control technology, partly replaced the traditional manual pesticide equipment which had the problems of dripping and extensive application of pesticide. Furthermore, it can make up for the lack of field management time caused by part-time operation. This form of outsourcing provides a solution to the problem that small farmers cannot reinvest in environment-friendly agricultural control equipment due to their limited economic income, and is beneficial to the sustainable development of agriculture.

According to the research conclusion, it can draw some countermeasures and suggestions:

First of all, we should continue to strengthen the outsourcing control organizations of rice pest and disease, and encourage them to have different enterprise roles. In particular, the government needs to pay attention to the development of cooperative prevention and control organizations. The service scope of outsourcing organization covers the village, which means that the lower the cost of information transmission of outsourcing prevention and control, the lower the transaction cost of farmers and outsourcing organizations to discuss prevention and control matters, and rice farmers are more likely to accept the new outsourcing prevention and control service mode.

Secondly, the government can encourage outsourcing organizations to improve their plant protection models to choose the advanced nature, improve the efficiency of control and prevention of machinery. Farmers adopt the form of outsourcing to replace the independent

control, in essence, it is to meet the needs of saving labor time in the field. Especially for the main migratory pests of rice (such as rice planthopper, rice leaf roller), there may be adult population transferred to adjacent plots to continue to harm, reciprocating spraying, the actual control effect is limited. Compared with the efficiency of household holding machinery, the comparative advantage of prevention and control efficiency adopted by outsourcing organizations is obvious, which can help farmers to achieve the purpose of saving labor and time. In particular, encourage aviation plant protection in the form of complex terrain to develop micro multi rotor plant protection UAV, in order to expand the appropriate scale of outsourcing services, and then expand the regional service scope of local township outsourcing organizations.

Finally, more stable non-agricultural jobs should be provided in cities and towns, and more standardized control agreements should be signed between prevention and control organizations and farmers to reduce the cost of supervision, so as to encourage the mechanical efficiency of outsourcing organizations to replace the field input of human capital, so as to realize the trusteeship service of rice control.

There are still some problems worthy of further study. Are there differences in control outsourcing behaviors among different crop types? What is the ecological effect of specific field? What is the comparative evaluation of outsourcing efficiency in different regions? We hope to find the answer in the follow-up study.

## Supporting information

**S1 File.**
(DTA)

## Acknowledgments

We gratefully thank the interviewed farmers who gave their time and resources to participate in the survey, and Mr. Ruifeng Guan, the master of Fujian Plant Quarantine and Protection Station, supported our investigation. Lastly, we thank Dr. Zhuo Chen and Dr. Shun Xiao for useful suggestions to this manuscript.

## Author Contributions

**Conceptualization:** Linping Wang.

**Data curation:** Liangmei Cai, Linping Wang.

**Funding acquisition:** Linping Wang.

**Methodology:** Liangmei Cai.

**Resources:** Linping Wang.

**Software:** Liangmei Cai.

**Supervision:** Linping Wang.

**Writing – original draft:** Liangmei Cai.

**Writing – review & editing:** Linping Wang.

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
