## [Decision Letter · Decision Letter 0]

15 May 2021

PONE-D-21-12763

Analysis on Outsourcing Service Behavior of Rice Pest and Disease Control Based on Heckman Selection Model

PLOS ONE

Dear Dr. LinPing Wang,

Thank you for submitting your manuscript to PLOS ONE. After careful consideration, we feel that it has merit but does not fully meet PLOS ONE’s publication criteria as it currently stands. Therefore, we invite you to submit a revised version of the manuscript that addresses the points raised during the review process.

We look forward to receiving your revised manuscript.

Kind regards,

László VASA, PhD

Academic Editor

PLOS ONE

Journal Requirements:

4. We note that Figure 1 in your submission contains map images which may be copyrighted. All PLOS content is published under the Creative Commons Attribution License (CC BY 4.0), which means that the manuscript, images, and Supporting Information files will be freely available online, and any third party is permitted to access, download, copy, distribute, and use these materials in any way, even commercially, with proper attribution. For these reasons, we cannot publish previously copyrighted maps or satellite images created using proprietary data, such as Google software (Google Maps, Street View, and Earth). For more information, see our copyright guidelines: http://journals.plos.org/plosone/s/licenses-and-copyright.

Reviewers' comments:

Reviewer's Responses to Questions

**Comments to the Author**

1. Is the manuscript technically sound, and do the data support the conclusions?

Reviewer #1: Partly

Reviewer #2: Yes

2. Has the statistical analysis been performed appropriately and rigorously? 

Reviewer #1: Yes

Reviewer #2: Yes

3. Have the authors made all data underlying the findings in their manuscript fully available?

Reviewer #1: Yes

Reviewer #2: Yes

4. Is the manuscript presented in an intelligible fashion and written in standard English?

Reviewer #1: Yes

Reviewer #2: Yes

5. Review Comments to the Author

Reviewer #1: The paper deals with a basically peripherical issue in general, which, however, seems to be important in its field, indeed. However, the topic is relatively narrow; 1) having a potential interest for it from the readers' side, 2) can contribute to the existing knowledge to its field.

Basically, in this paper, the appropriate methodology is used to investigate the research questions. This methodology supports the results, and the conclusions are based on the results.

However, I have problems with the structure, which is quite unclear and partially with the contents.

- The introduction is not comprehensive; the research questions are not set appropriately, and the context is not explained.

- Chapters 2, 3, and 4 should be one chapter called "Material and Methods." However, in this case, the analysis part is too short.

I recommend writing more in the analysis chapter - inputs the authors have enough based in their research.

- I could not find any literature review in the paper; some resources are cited in the introduction and methodology chapters; however, the actual literature review is missing. Therefore, I recommend writing an analytical, critical, and comprehensive literature review chapter.

- there are no limitations of the research indicated.

Reviewer #2: The Manuscript "Analysis on Outsourcing Service Behavior of Rice Pest and Disease Control Based on

Heckman Selection Model" addresses an interesting, important topic along developing field crop (rice) production, specifically plant protection in an exciting socio-economic and "farmers behavior" context in Fujian Province, China.

The data collection in selected regions, counties and the number of farmers is appropriate for processing and coming to conclusions. The pest control/management and the potential outsourcing is very well introduced, discussed in the Manuscript. The conclusions are well phrased and imprortant for policy development purposes as well.

Three remarks:

- the category "pest" usually covers all harmfull organisms, incl. vuruses, fungi, animals, etc. Thus pests and diseases in the text may be corrected for arthropod pests and diseases.

- in addition to outsourcing of "pest control" service, a broader outsourcing of "pest management" is more appropriate approach.

- the Authors may add some sentence to the discussion/conclusion on how the outsourcing service may contribute to broader IPM implementation, sustainable production, farming development.

6. PLOS authors have the option to publish the peer review history of their article (what does this mean?). If published, this will include your full peer review and any attached files.

Reviewer #1: No

Reviewer #2: No

---

## [Author Response · Author response to Decision Letter 0]

18 Jun 2021

Dear László VASA, PhD and Reviewers:

 Thank you for your letter and for the reviewers’comments concerning our manuscript entitled “Analysis on Outsourcing Service Behavior of Rice Pest and Disease Control Based on Heckman Selection Model-- A Case Study of Ten Counties in Fujian Province” (ID: PONE-D-21-12763). Those comments are all valuable and very helpful for revising and improving our paper, as well as the important guiding significance to our researches. We have studied comments carefully and have made correction which we hope meet with approval. 

A list of changes and responses to reviews are as follows.

List of Action

(1) We have amended paper format according to the PLOS ONE style templates.

(2) The corresponding author used the ORCID id.

(3) We amended the title on the online submission form.

(4) We removed the survey area map and replaced it with a place name table.

Responds to the reviewer’s comments:

Reviewer #1: 

Special thanks to you for your good comments. 

（1）Response to comment: “The introduction is not comprehensive; the research questions are not set appropriately, and the context is not explained.”

Response: In the introduction, we tried to amend the language expression to make it more logical. And we also added the performance of China's agricultural outsourcing service into the literature review for explaining the context.

（2）Response to comment: “Chapters 2, 3, and 4 should be one chapter called "Material and Methods.”, and write more in the analysis chapter”

Response: The article merges the chapter 2,3,4 in the original manuscript into the new chapter 3 "Material and Methods". In the new manuscript, the chapter 4 “Model estimation results” presents the empirical results of the model and the results are analyzed.

（3）Response to comment: write an analytical, critical, and comprehensive literature review chapter.

Response: Thanks to the reviewers’ comment, we have added a literature review chapter, which contains the following parts： “2.1 Theoretical basis of agricultural outsourcing service”； “2.2 Study on the influencing factors of agricultural outsourcing services participation”；“2.3 The influence of agricultural outsourcing services on other production factors”；“2.4 Research gap”(Page3-Page5)

（4）Response to comment: - there are no limitations of the research indicated.

Response: In the ending of chapter 5 “Conclusion and suggestion” ，we put forward some problems worthy of further study. 

Reviewer #2: 

Special thanks to you for your good comments. 

（1）Response to comment: “the category "pest" usually covers all harmful organisms, incl. viruses, fungi, animals, etc. Thus, pests and diseases in the text may be corrected for arthropod pests and diseases.”

Response: It is really true as Reviewer suggested that the definition of “pest”is too broad. So we have defined and annotated "pest" according to the reviewer’s suggestion in page 1.

（2）Response to comment: “in addition to outsourcing of "pest control" service, a broader outsourcing of "pest management" is more appropriate approach.”

Response: Thank you for your suggestions. According to some scholars,“pest control “maybe used more frequently (such as “Chen Pin, Zhong Funing, Sun Dingqiang. Farming Time Delay, Yield Loss, and Outsourcing Service Utilization under Labor Shortages: Taking Rice Pest Control as An Example [C], Chinese Conference on Agricultural Forestry Economics and Management (2017CAFEM)”and “Ying R Y,Xu B.Effects of Specialized Crop Pest and Disease Control Services on The Intensity of Pesticide Application[J].Population Resources and Environment in China,2017(08):90-97.”). So we more likely to use “pest control”.

（3）Response to comment: “The Authors may add some sentence to the discussion/conclusion on how the outsourcing service may contribute to broader IPM implementation, sustainable production, farming development.”

Response: In the chapter5“Conclusion and suggestion”(Page.18）,we added further explanations about how the outsourcing service may contribute to broader IPM implementation, sustainable production, farming development.

 We tried our best to improve the manuscript and made some changes in the manuscript. These changes will not influence the content and framework of the paper. And here we did list the changes and marked in red/blue in revised paper.

We appreciate for László VASA, PhD and reviewers warm work earnestly and hope that the correction will meet with approval.

 Once again, thank you very much for your comments and suggestions.

 LinPing Wang

 06.14.2021

---

## [Decision Letter · Decision Letter 1]

5 Jul 2021

Analysis on Outsourcing Service Behavior of Rice Pest and Disease Control Based on Heckman Selection Model-- A Case Study of Ten Counties in Fujian Province

PONE-D-21-12763R1

Dear Dr. LinPing Wang,

We’re pleased to inform you that your manuscript has been judged scientifically suitable for publication and will be formally accepted for publication once it meets all outstanding technical requirements.

Kind regards,

László Vasa, PhD

Academic Editor

PLOS ONE

Additional Editor Comments (optional):

Reviewers' comments:

Reviewer's Responses to Questions

**Comments to the Author**

1. If the authors have adequately addressed your comments raised in a previous round of review and you feel that this manuscript is now acceptable for publication, you may indicate that here to bypass the “Comments to the Author” section, enter your conflict of interest statement in the “Confidential to Editor” section, and submit your "Accept" recommendation.

Reviewer #1: All comments have been addressed

Reviewer #2: All comments have been addressed

2. Is the manuscript technically sound, and do the data support the conclusions?

Reviewer #1: Yes

Reviewer #2: Yes

3. Has the statistical analysis been performed appropriately and rigorously? 

Reviewer #1: Yes

Reviewer #2: Yes

4. Have the authors made all data underlying the findings in their manuscript fully available?

Reviewer #1: Yes

Reviewer #2: Yes

5. Is the manuscript presented in an intelligible fashion and written in standard English?

Reviewer #1: Yes

Reviewer #2: Yes

6. Review Comments to the Author

Reviewer #1: The authors accepted my comments and made the necessary improvements. Based on this latest version, I can accept it for publication.

Reviewer #2: The manuscripzt went through important changes, modifications that make it more clear, explicit and meaningful.

One remark where Reviewer raises reservation is the use of "pest control" instead of "pest management".

7. PLOS authors have the option to publish the peer review history of their article (what does this mean?). If published, this will include your full peer review and any attached files.

Reviewer #1: No

Reviewer #2: No

---

## [Editor Report · Acceptance letter]

9 Jul 2021

PONE-D-21-12763R1 

Analysis on Outsourcing Service Behavior of Rice Pest and Disease Control Based on Heckman Selection Model
-- A Case Study of Ten Counties in Fujian Province 

Dear Dr. Wang:

I'm pleased to inform you that your manuscript has been deemed suitable for publication in PLOS ONE. Congratulations! Your manuscript is now with our production department. 

Kind regards, 

on behalf of

Prof. Dr. László Vasa 

Academic Editor

PLOS ONE